# Genetic and Pathogenic Characterization of Avian Influenza Virus in Migratory Birds between 2015 and 2019 in Central China

Zhongzi Yao,[a,b] Huabin Zheng,[a,b] Jiasong Xiong,[a,b] Liping Ma,[a,b] Rui Gui,[a,b] Gongliang Zhu,[c] Yong Li,[c] Guoxiang Yang,[c] Guang Chen,[c] Jun Zhang,[c] Quanjiao Chen[a]

[a]CAS Key Laboratory of Special Pathogens and Biosafety, Wuhan Institute of Virology, Center for Biosafety Mega-Science, CAS Center for Influenza Research and Early Warning, Chinese Academy of Sciences, Wuhan, China
[b]University of Chinese Academy of Sciences, Beijing, China
[c]The Monitoring Center of Wildlife Diseases and Resource of Hubei Province, Wuhan, China

Zhongzi Yao and Huabin Zheng contributed equally to this work. Author order was determined on the basis of seniority.

**ABSTRACT** Active surveillance of avian influenza virus (AIV) in wetlands and lakes is important for exploring the gene pool in wild birds. Through active surveillance from 2015 through 2019, 10,900 samples from wild birds in central China were collected, and 89 AIVs were isolated, including 2 subtypes of highly pathogenic AIV and 12 of low-pathogenic AIV; H9N2 and H6Ny were the dominant subtypes. Phylogenetic analysis of the isolates demonstrated that extensive intersubtype reassortments and frequent intercontinental gene exchange occurred in AIVs. AIV gene segments persistently circulated in several migration seasons, but interseasonal persistence of the whole genome was rare. The whole genomes of one H6N6 and polymerase basic 2 (PB2), polymerase acidic (PA), hemagglutinin (HA), neuraminidase (NA), M, and nonstructural (NS) genes of one H9N2 virus were found to be of poultry origin, suggesting a spillover of AIVs from poultry to wild birds. Importantly, one H9N2 virus only bound to human-type receptor, and one H1N1, four H6, and seven H9N2 viruses possessed dual receptor-binding capacity. Nineteen of 20 representative viruses tested could replicate in the lungs of mice without preadaptation, which poses a clear threat of infection in humans. Together, our study highlights the need for intensive AIV surveillance.

**IMPORTANCE** Influenza virus surveillance in wild birds plays an important role in the early recognition and control of the virus. However, the AIV gene pool in wild birds in central China along the East Asian-Australasian flyway has not been well studied. Here, we conducted a 5-year AIV active surveillance in this region. Our data revealed the long-term circulation and prevalence of AIVs in wild birds in central China, and we observed that intercontinental gene exchange of AIVs is more frequent and continuous than previously thought. Spillover events from poultry to wild bird were observed in H6 and H9 viruses. In addition, in 20 representative viruses, 12 viruses could bind human-type receptors, and 19 viruses could replicate in mice without preadaption. Our work highlights the potential threat of wild bird AIVs to public health.

**KEYWORDS** avian influenza virus, pathogenic, reassortment, viral surveillance, wild birds

Address correspondence to Quanjiao Chen, chenqj@wh.iov.cn.

The authors declare no conflict of interest.

Migratory birds are the natural source and reservoir of avian influenza viruses (AIVs) and are capable of harboring all subtypes of AIVs (1, 2). Highly pathogenic AIV (HPAIV) and low-pathogenic AIV (LPAIV) can spread globally via bird migration and pose a threat to the poultry industry and global human health (3). In January 2014, an outbreak of H5N8 HPAIV in poultry was detected in South Korea; shortly after, the virus

spread rapidly worldwide by migratory wild birds, which caused disease outbreaks in poultry in Asia, Europe, and North America (4–6). The potential threat of LPAIV should not be underestimated as well. LPAIVs in wild birds contribute to the genetic diversity of AIVs in poultry, and some isolates have even crossed the species barrier to infect humans. The N9 gene of the H7N9 subtype, which caused 1,567 infections in humans and 608 deaths as of 7 February 2020, is considered to have originated from wild birds (7). In China, H10N8 viruses, which have been detected in three humans since December 2013, harbored the surface glycoproteins genes originating from wild birds and internal genes originating from AIVs circulating among domestic poultry (8).

From 2007 through 2011, the geographical separation of AIV host species presumably separated the AIV gene pool into independently evolving Eurasian and American lineages, although reassortment between these lineages occurred on occasion (9–11). Since 2011, intercontinental spread of the virus, whether of North American or Eurasian lineage, has been reported (12–14). Understanding the potential for genetic interchange of AIV between hemispheres is critical to limiting spread and mitigating the effects of AIV on human and animal health.

In this study, we conducted active AIV surveillance of wild birds in central China along the East Asian-Australasian (EA) flyway. The EA flyway stretches from the Russian Far East and Alaska, southward through East Asia and Southeast Asia, to Australia and New Zealand, and it encompasses 22 countries. The EA flyway is home to over 50 million migratory waterbirds, including over 250 different populations, along with 36 globally threatened species and 19 near-threatened species. During migration, waterbirds rely on a system of highly productive wetlands to rest and feed, building up sufficient energy to fuel the next phase of their journey (15). Sampling sites in this study were mainly located within the Wang Lake Wetland Reserve, Poyang Lake National Nature Reserve, and East Dongting Lake National Nature Reserve, which are the wintering and stopover sites for thousands of migratory birds that fly along the EA flyway. In the winter and spring, migratory waterfowl aggregate at these wetlands, making them ideal places for conducting research on the AIV gene pool in wild birds. Here, we describe the results of AIV surveillance of wild birds in central China over a period of 5 years (2015 to 2019). Our results clearly reveal the viral prevalence, genetic diversity, receptor binding specificity, and replicative ability in mammals of AIVs circulating in wild birds in central China.

## RESULTS

**AIV distribution in migratory birds in central China from 2015 to 2019.** Between 2015 and 2019, during the migration season (from November to March of the following year), we collected 10,910 fecal samples of migratory birds around lakes and wetlands in Hunan, Hubei, and Jiangxi provinces (central China) (Fig. 1A). Eighty-nine AIVs were isolated from the collected samples (see Table S1 in the supplemental material). The AIV positivity rates of each migration season (Fig. 1B) varied from 0.08% (2017 to 2018) to 1.38% (2018 to 2019) (Table S1).

Twelve hemagglutinin (HA)-neuraminidase (NA) subtype combinations of influenza A virus, including five HA (H1, H5, H6, H9, and H10) and six NA (N1, N2, and N5 to N8) subtypes, were isolated (Fig. 1C). The most prevalent subtype was H9N2 (29/89, 32.58%). However, the most prevalent HA was H6 (44/89,49.44%), combined with four NAs (N1, N2, N5, and N6) (Table S1). In the four consecutive migratory seasons, the prevalent AIV subtype in migratory birds fluctuated. The dominant subtypes were H6N2 (17/23), H9N2 (19/46), and H9N2 (9/19) in the 2015 to 2016, 2016 to 2017, and 2018 to 2019 seasons, respectively. Since only one strain was isolated in the 2017 to 2018 season, data analysis was not possible (Table S1). Notably, the H9 and H6 subtypes were all isolated in the above-mentioned three seasons. The dominant H6 subtype switched from H6N2/H6N1 to H6N5 in the 2018 to 2019 season.

**Phylogenetics and genotyping analysis of AIVs.** To understand the genetic distribution of the 89 isolated AIVs, maximum-likelihood trees of HA and NA genes were constructed (Fig. 2A to E; Fig. S1). Phylogenetic analysis showed that the HA genes of H1 and H10 subtypes are clustered with those of the other H1Ny and H10Ny viruses

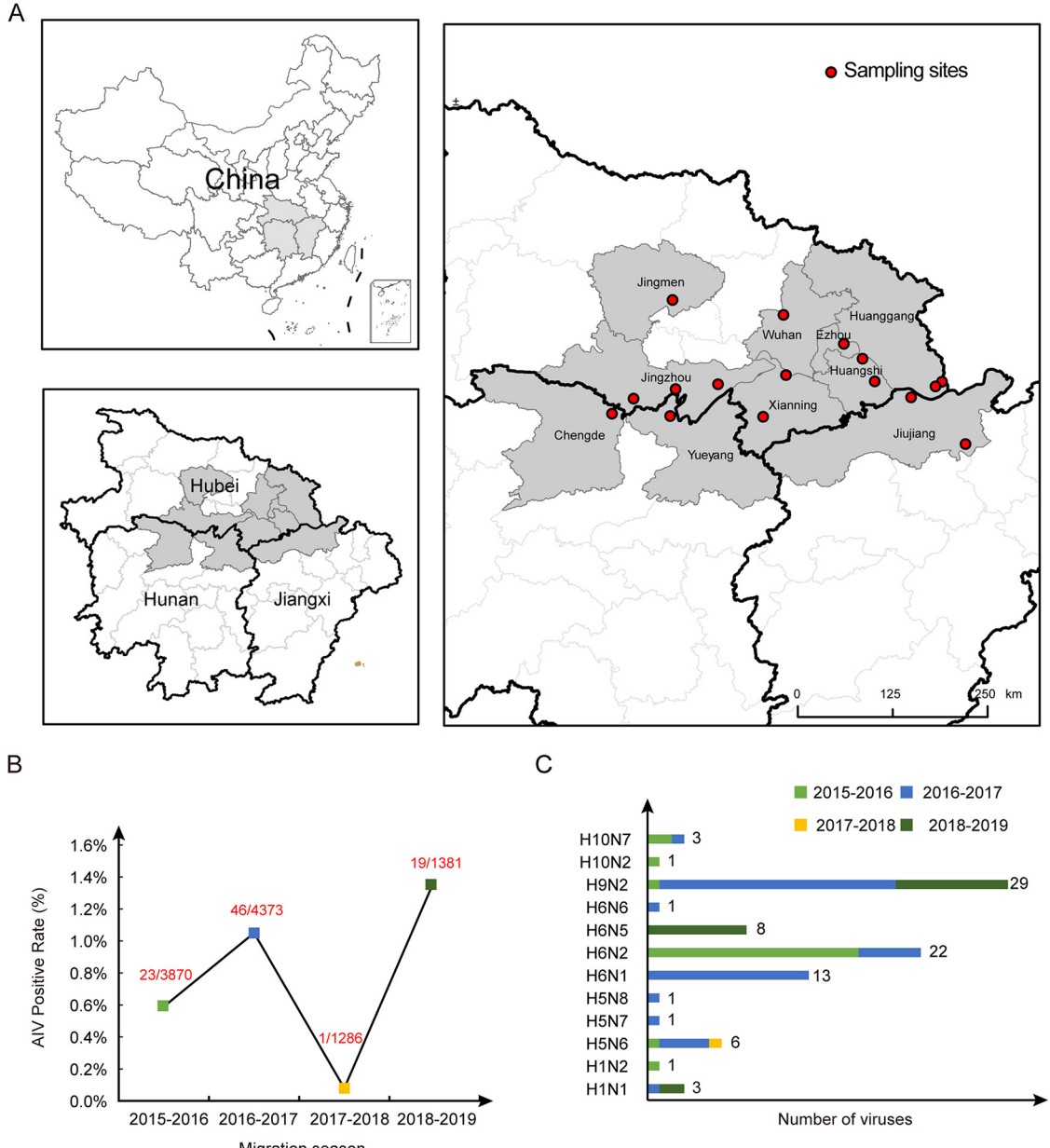

**FIG 1** Map of sampling sites and distribution of AIVs in wild birds along the EA flyway. (A) Geographical locations of the sampling sites in central China. The map was created in ArcGIS 10.2 software (ESRI Inc., Redlands, CA, USA). (B) AIV-positive ratios during four migration seasons from 2015 to 2019. The number in red means number of AIV-positive samples/number of total samples. (C) Subtype proportions of the AIVs isolated in four migration seasons.

derived from wild birds in Eurasia, respectively (Fig. 2A and E; Fig. S1). In the H5 phylogenetic tree, five H5N6 viruses belong to clade 2.3.4.4h, and one H5N6 virus belongs to clade 2.3.4.4e. The H5N8 virus belongs to clade 2.3.4.4b. The H5N7 virus falls into the LPAI lineage (Fig. 2B). The phylogenetic trees of the H6 and H9 genes were separated into the North American and Eurasian lineages. All HA genes of H6N1, H6N2, and H6N6 viruses clustered into the Eurasian lineage, and those of H6N5 viruses belonged to North American lineages (Fig. 2C; Fig. S1). HA genes of 29 H9N2 viruses in this study could be divided into 3 groups. HA genes of seven viruses belong to the Eurasian wild bird AIV gene pool, and one H9N2 virus was clustered together with H9 viruses circulating in domestic poultry. HA genes of the remaining 21 H9N2 viruses were clustered into the North American lineage, with nucleotide identities ranging from 99.5 to 100%

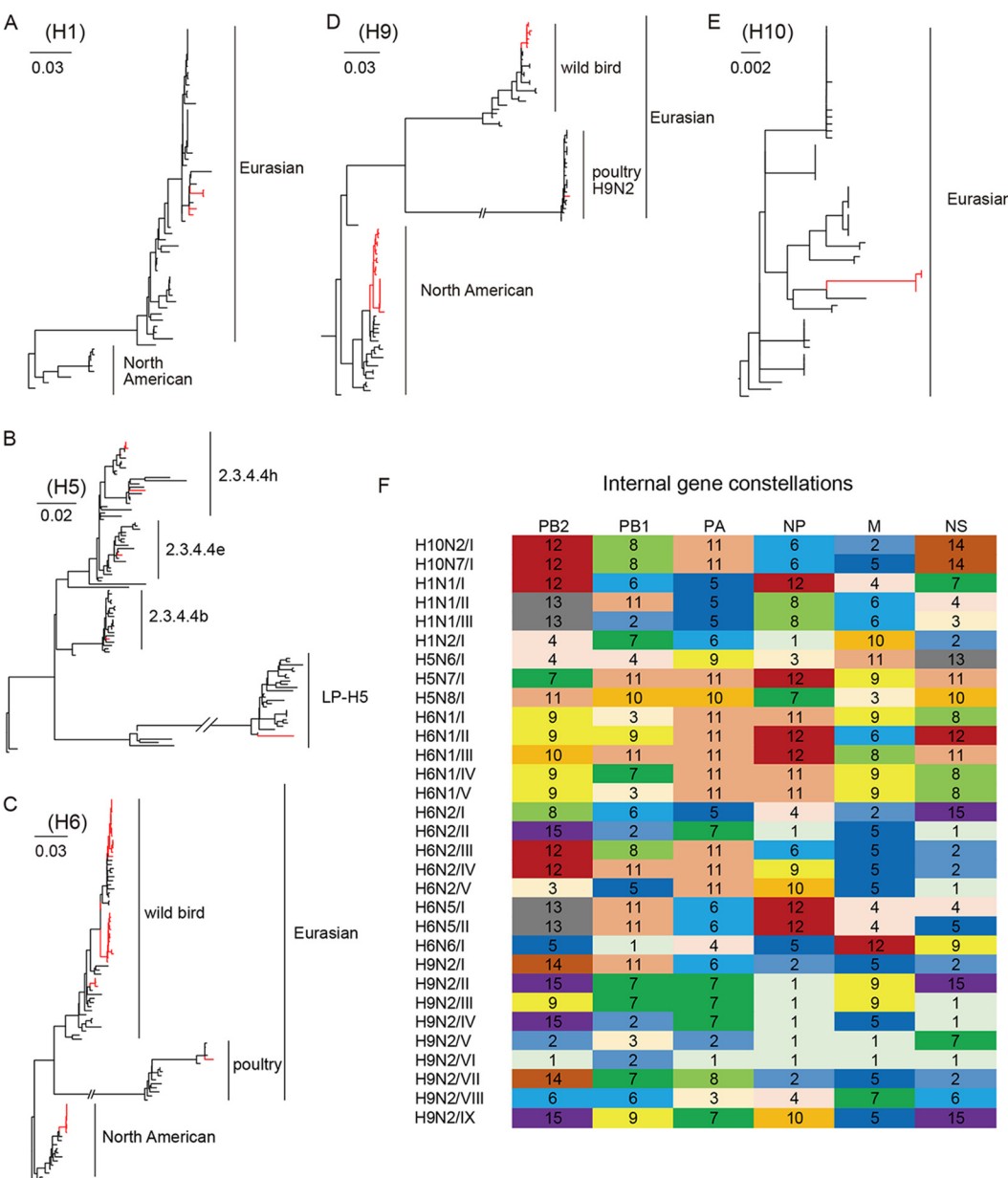

**FIG 2** Phylogenetic and genotyping analysis of 89 AIVs. Phylogenetic and genotyping analysis of 89 avian influenza viruses (AIVs). (A to E) Phylogenetic trees of the H1 (A), H5 (B), H6 (C), H9 (D), and H10 (E) gene segments. The WHO classification was used to describe the subclades of clade 2.3.4.4 HPH5 viruses. All trees are midpoint rooted for clarity. Sequences identified in this study are marked in red. Large versions of these phylogenetic trees are provided in Fig. S1 in the supplemental material. (F) Gene constellations of internal genes. The gene constellations were generated based on maximum-likelihood (ML) trees for each internal gene (shown in Fig. S2). Colors and numbers in each column depict groups (defined by strong bootstrap support [>80%] and similarity of different clades [<98%]) of the indicated gene segment on the phylogenetic tree.

(Fig. 2D; Fig. S1). For NA genes, phylogenetic analysis showed that the N1, N5, and N7 genes of AIVs in this study were closely related to those of viruses circulating among wild birds in Eurasia. NA genes of three H6N2 viruses and five H9N2 viruses belong to the North American lineage. The N2 gene of one H9N2 virus was closely related to H9 viruses circulating in poultry. N2 genes of remaining HyN2 viruses belong to the Eurasian wild bird AIV gene pool. NA genes of H5N8 viruses are clustered with those of HP H5N8 viruses circulating in Asia and Africa from 2016 through 2017 (Fig. S1).

We performed phylogenetic analyses of internal genes and classified them into different groups according to tree topology and bootstrap values (>80%) (Fig. 2F; Fig. S2).

According to the genotyping, all 89 AIVs were assigned to 31 distinct genotypes (Table S2). Both H6N1 and H6N2 viruses were divided into five genotypes. H9N2 viruses, which showed the highest diversity of gene distribution, were divided into nine different genotypes. Both H5N6 ($n = 6$) and H10N7 ($n = 3$) viruses only had one internal gene constellation. No virus with a different subtype shared the same gene distribution, but the polymerase basic 2 (PB2), PB1, polymerase acidic (PA), nucleoprotein (NP), and movement protein (MP) genes of H10N7/I and H6N2/III viruses clustered together in phylogenetic trees. These results indicate that frequent and complex reassortment events occurred within the same subtype viruses and among different subtype viruses.

The per-site diversity for each internal gene segment was also calculated (Fig. S3). Among the five HA subtypes, the lowest diversity was observed in the H10 subtype, consistent with the gene distribution results (Fig. 2F; Fig. S3). Among six internal genes, M was the most conserved segment in all subtypes, and the nonstructural (NS) gene segments of H1, H6, and H9 exhibited high diversity, which is compatible with the deep A and B allelic polymorphism in this segment.

**Persistent circulation of AIV genotypes and gene segments.** To test the persistent interseasonal circulation of viruses among wild birds, the distribution of genotypes in each migration season was analyzed (Fig. 3A). Except for H5N6/I, which appeared in three seasons (2015 to 2016, 2016 to 2018, and 2017 to 2018), the other genotypes appeared in only one migration season. Internal genes which clustered into the same gene group (Fig. S2) as those in other migration seasons are defined as persistence sequences. The number of persistence sequences was also calculated (Fig. 3B). Two groups of PB2, six groups of PB1, five groups of NP and NS, and four groups of PA and M were detected in different migration seasons. The highest number of persistence sequences was observed in M group 5. Taken together, the results indicated that internal genes may have circulated for years in the wild bird population, but whole-genome persistence is rare.

**AIVs of the North American lineage invade Eurasia.** Our phylogenetic analyses revealed that a large number of viruses contained gene segments of the North American lineage (Fig. S1 and S2). A total of 121 segments from 48 isolates of 5 subtypes had been derived from North America, including the PB1 gene of 1 H1N1 strain; the NP gene of 1 H1N2 virus; the NA genes of 3 H6N2 viruses; the PB1, NP, and NS genes of 4 H6N2 strains; the PB1 genes of 9 H6N1 strains; the HA genes of 8 H6N5 strains; the whole genome except the NS genes of 3 H9N2 viruses; the HA and NA genes of 9 H9N2 viruses; the NP and NA genes of 1 H9N2 virus; the HA, NP, NA, and NS genes of 5 H9N2 viruses; the PB1, HA, NP, NA, and NS genes of 2 H9N2 strains; and the entire genome of 2 H9N2 viruses (Fig. 4). In summary, almost half (48/89, 53.93%) of the isolated viruses harbored genes derived from North America.

Gene invasion was especially evident in the H9N2 virus, and 75.86% (22/29) of H9N2 viruses carried the North American lineage genes. The 22 viruses were isolated in 2 migration seasons (2016 to 2017 and 2018 to 2019). Among them, the entire genomes of two viruses of the North American lineage were isolated in the 2016 to 2017 season. The Markov chain Monte Carlo (MCMC) analysis revealed that these 22 viruses are evolutionarily related. To facilitate the description of intercontinental reassortment, the 22 H9N2 viruses with the North American lineage genes were divided into six invading patterns according to which gene segments derived from North American lineage (named NAm1 to NAm6) (Fig. 4; Fig. S4). Two North American-derived viruses belonged to NAm6, and for three viruses, the whole genomes except the NS gene were NAm5. Evolutionary analysis showed that the HA genes of 21 viruses (except the NAm2 virus, the HA segment of which belongs to the Eurasian lineage) clustered together, and NP of 13 viruses (except 9 NAm1 viruses, the NP segments of which belong to the Eurasian lineage) clustered together, indicating a common ancestor. All the North American-derived genes of NAm3 and NAm4 (the HA, NP, and NS genes of NAm3 and the PB1, HA, NP, and NS genes of NAm4) were clustered with those of the NAm6 isolates, suggesting that NAm3 and NAm4 viruses underwent reassortment between NAm6 and Eurasian lineages. The HA and NP genes of the NAm5

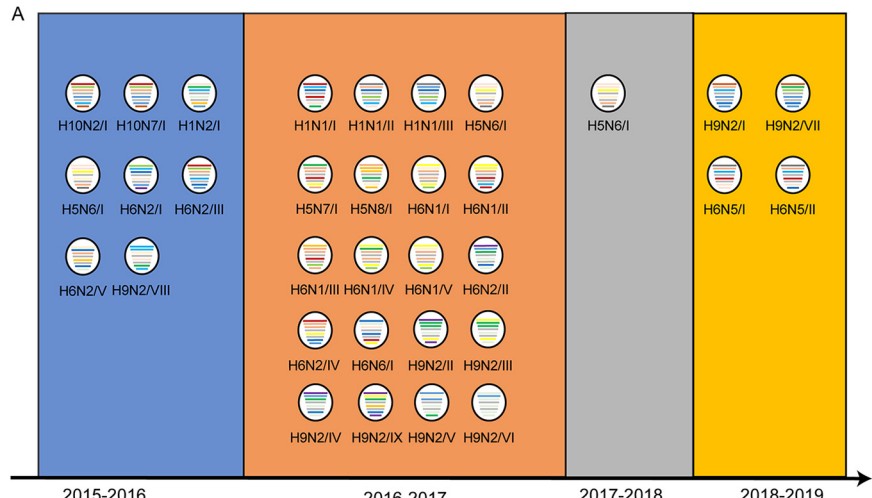

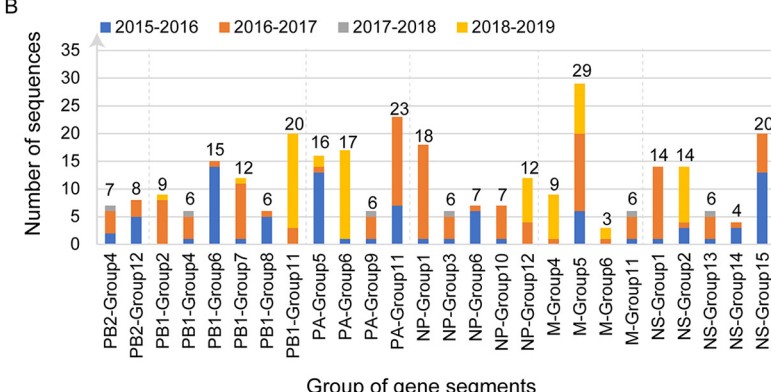

FIG 3 Analysis of interseasonal persistence. (A) Genotype distribution of viruses isolated in four migration seasons. The eight gene segments are indicated by horizontal bars within the ovals (from top to bottom, PB2, PB1, PA, HA, NP, NA, M, and NS). The color of the internal gene segment represents a distinct clade related to that depicted in Fig. 1F and Fig. S2 in the supplemental material. (B) Number of persistent internal gene segments. Gene segments derived from strains isolated in different migration seasons but clustered into the same gene group (detailed in Fig. S2) are defined as persistent gene segments. Different colors represent the distinct sampling season.

viruses were clustered with those of NAm6 viruses, but the PB2, PB1, PA, NA, and MP genes of the NAm5 viruses formed a distinct invading subclade, being closely related to those of an H9N2 strain from Alaska (A/cackling/goose/Alaska/UGAI15-3075/2015 [Alaska/3075]).

The PB1 genes of nine H6N1 viruses were close to those of NAm5 viruses, while the PB1, NP, and NS genes of four H6N2 viruses and the PB1 gene of one H1N1 virus clustered with those of NAm6 viruses (Fig. S2). These findings suggest that intercontinental reassortment events occurred among subtypes.

**Spillover of viruses from poultry to wild birds.** Phylogenetic analysis showed that all genes of the H6N6 virus (A/wild bird/Hubei/01.24_FHC172-1/2017 [H6N6], FHC172-1) are closely related to those of viruses that were isolated from poultry (Fig. 2; Fig. S1 and S2). This suggests that AIVs from poultry are being introduced into wild birds. The PB1 and NP genes of one H9N2 virus (A/wild bird/Hubei/03.09F3-133-2/2016 [F3-133-2]; the abbreviation of each strain has been listed in Table S2) clustered with those of viruses isolated from wild birds, but the remaining genes (PB2, PA, HA, NA, M, and NS) clustered with those from poultry. This suggests that viruses derived from poultry reassorted with those circulating in wild birds and produced the reassortant strain F3-133-2.

**Molecular characterization.** To evaluate the threat of AIVs carried by wild birds to human health, we characterized the zoonotic potential of the isolated viruses, identified

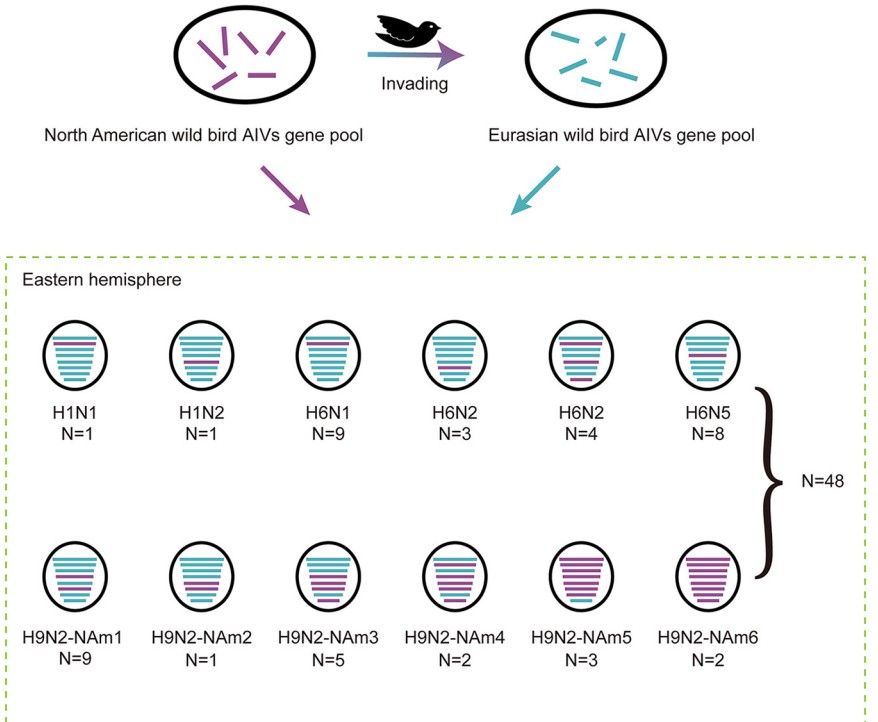

**FIG 4** Intercontinental invading pattern of AIVs. The eight gene segments are indicated by horizontal bars within the ovals (from top to bottom, PB2, PB1, PA, HA, NP, NA, M, and NS). All gene segments belong to the Eurasian lineage are colored green. Genes derived from the North American lineage are colored purple.

any adaptive mutations to mammals or humans, and mapped the key amino acids associated with enhanced virulence and resistance to antiviral drugs.

For six H5N6 viruses and one H5N8 virus, multiple basic amino acids (RERRRKR) were found to be carried at the HA cleavage site, which is a classic HPAI feature (Table S2). All H5N6 isolates belong to clade 2.3.4.4, as the human-infecting H5N6 exhibited some human-like signatures, including PA-A404S, HA-I155T, and T160A (Table 1) (16–18). The HA-155T and T160A mutations within the receptor-binding site increase the binding preference for human-type receptors and transmissibility in guinea pigs (19). For other subtypes, PB2-588V (one H1N2 and two H9N2), PA-356R (one H9N2), PA-382D (eight H6N5 and eight H9N2), and PA-409N (one H6N6 and one H9N2) mutations were detected at these sites (Table 1; Table S2). These mutations are related to increased polymerase activity and viral replication in mammals (20–22). The HA-I155T mutation was also detected in HA of all 3 H1N1, 1 H1N2, and 29 H9N2 viruses. Additionally, F3-133-2/2016 (H9N2) had a 3-amino-acid-long deletion in the NA stalk and had an A325S substitution at the HA-P5 cleavage site (Table 1; Table S2) (23). The NA stalk deletion combined with the A325S mutation in HA can increase HA cleavage efficiency. H5N7 and H5N8, 2 H6N1, and 18 H6N2 viruses possess the T160A substitution in HA. NS-42S was detected in 3 H1N1, 6 H5N6, 1 H5N7, 1 H5N8, 13 H6N1, 13 H6N2, 8 H6N5, 1 H6N6, 11 H9N2, 1 H10N2, and 3 H10N7 viruses. This mutation can increase virulence in mice (24). M2-S31N mutations were detected in H6N1, H6N5, and H9N2 viruses (one each); this mutation confers viral resistance to M2 blocker antiviral drugs, such as amantadine (25). No viruses carried mutations related to NA inhibitor resistance.

**Receptor-binding properties.** The molecular characterization results indicated that some viral proteins had identical amino acids at their receptor-binding sites and could partially bind to human-type receptors. Therefore, a receptor-binding test was performed to verify this observation using 20 representative strains, including H9N2 (*n* = 8), H6N2 (*n* = 4), H6N1 (*n* = 3), H1N1 (*n* = 1), H6N5 (*n* = 2), H6N6 (*n* = 1), and

**TABLE 1** Key amino acid residues of AIVs in this study[a]

| Function | Site | No. of mutations in: | | | | | | | | | | | |
|---|---|---|---|---|---|---|---|---|---|---|---|---|---|
| | | H1N1 (n = 3) | H1N2 (n = 1) | H5N6 (n = 6) | H5N7 (n = 1) | H5N8 (n = 1) | H6N1 (n = 13) | H6N2 (n = 22) | H6N5 (n = 9) | H6N6 (n = 1) | H9N2 (n = 29) | H10N2 (n = 1) | H10N7 (n = 3) |
| Enhanced replication of AIVs in mammals | PB2-588V | 0 | 1 | 6 | 0 | 0 | 0 | 0 | 0 | 0 | 2 | 0 | 0 |
| | PA-57Q | 0 | 0 | 0 | 0 | 0 | 0 | 0 | 0 | 0 | 2 | 0 | 0 |
| | PA-356R | 0 | 0 | 0 | 0 | 0 | 0 | 0 | 1 | 0 | 1 | 0 | 0 |
| | PA-382D | 0 | 0 | 0 | 0 | 0 | 0 | 0 | 8 | 0 | 8 | 0 | 0 |
| | PA-404S | 0 | 0 | 5 | 0 | 0 | 0 | 0 | 0 | 1 | 0 | 0 | 0 |
| | PA-409N | 0 | 0 | 0 | 0 | 0 | 0 | 0 | 0 | 1 | 1 | 0 | 0 |
| Avian-to-human receptor-binding adaptation | HA-155T | 3 | 1 | 6 | 0 | 0 | 0 | 0 | 0 | 0 | 29 | 0 | 0 |
| | HA-160A | 0 | 0 | 6 | 1 | 1 | 2 | 18 | 0 | 0 | 0 | 0 | 0 |
| | HA-183N | 0 | 0 | 0 | 0 | 0 | 0 | 0 | 0 | 0 | 1 | 0 | 0 |
| | HA-226L | 0 | 0 | 0 | 0 | 0 | 0 | 0 | 0 | 0 | 1 | 0 | 0 |
| | HA-228S | 0 | 0 | 0 | 0 | 0 | 0 | 0 | 0 | 1 | 0 | 0 | 0 |
| | HA-325S | 3 | 1 | 0 | 0 | 0 | 0 | 0 | 0 | 0 | 1 | 0 | 0 |
| | NA stalk deletion[b] | 0 | 0 | 0 | 0 | 0 | 0 | 0 | 0 | 0 | 1 | 0 | 0 |
| Increased virulence | NS-42S | 3 | 0 | 6 | 1 | 1 | 13 | 13 | 8 | 1 | 11 | 1 | 3 |
| | M2-27I | 0 | 0 | 0 | 0 | 0 | 0 | 0 | 0 | 1 | 0 | 0 | 0 |
| Resistance to antiviral drugs | M2-31N | 0 | 0 | 0 | 0 | 0 | 1 | 0 | 1 | 0 | 1 | 0 | 0 |

[a]The amino acids of HA and NA using H3 numbering and N2 numbering, respectively.
[b]Three-amino-acid-long deletion in the NA stalk at positions 63 to 65.

H10N7 (n = 1). All H9N2 viruses could bind to $\alpha$-2,6-linked glycans (Fig. 5). An H9N2 virus (F3-133-2/2016), of which the HA gene is closely related to viruses circulating in poultry, only bound to human-type receptors. F3-133-2/2016 (H9N2) harbored multiple mutations (such as HA-155T, HA-183N, HA-222L, and HA-226L) that contributed to human-type receptor-binding properties (Table S2). Seven other H9N2 viruses with 226Q possessed dual receptor-binding ability, although their affinity for $\alpha$-2,3-linked glycans was higher. For the H6 subtype, an H6N1 virus (WHF2W3-1/2016) and three H6N2 viruses (DTHF2-96-2/2016, JZJLB728-1/2016, and DTHF42-1/2017) exhibited both avian- and human-type receptor-binding ability (Fig. 5). All H6N5, H6N6, and H10N7 viruses only bound to avian-type receptors. The H1N1 virus BCF13-1/2019 (H1N1) with HA-155T substitutions could bind to both avian- and human-type receptors with no significant difference in affinity.

**Replication and pathogenicity of AIVs in mice.** To assess the potential public risks of the epidemic AIVs, the 20 representative viruses mentioned above were used for testing viral replication and pathogenicity in mice. In each experimental group, three 6-week-old female BALB/c mice were inoculated with $10^6$ 50% egg-infective dose ($EID_{50}$) of virus. The nasal turbinate, heart, lung, spleen, kidney, and brain were collected from three animals per group at 3 days postinfection (dpi). The viral titer was tested in MDCK cells using the PFU assay. Five animals were monitored for 14 days, and body weight was measured every day (Fig. 6A to C). Except for one H9N2 virus (YJZD55/2017), all other viruses could be detected in the lungs of infected mice, with mean titers varying from 1.87 to 4.4 $\log_{10}$ PFU/mL (Table 2). In groups infected with one H10N7 (DTHF3-17-1) and two H6N1 (WHF2W3-1 and FHC196-1) viruses, virus could be detected in the nasal turbinate, with mean titers varying from 1.7 to 2.17 ($\log_{10}$ PFU/mL) (Fig. 6D; Table S2). The maximum body weight loss was 13.2%, caused by inoculation with the FHC196-1/2017 (H6N1) virus. Mice infected with JJCHBCT20/2019 (H6N5) and FHC172 (H6N6) viruses had 10.3% and 11.0% maximum weight loss, respectively. The body weight of mice inoculated with the remaining AIVs only slightly decreased during the observation period. Except for body weight loss, no other clinical symptoms were observed in any group.

No AIV was detected in other mouse organs, including the heart, spleen, kidney, and brain. The results reveal that a considerable number of AIVs carried by wild birds can replicate in mammals without preadaptation and cause mild weight loss in mice.

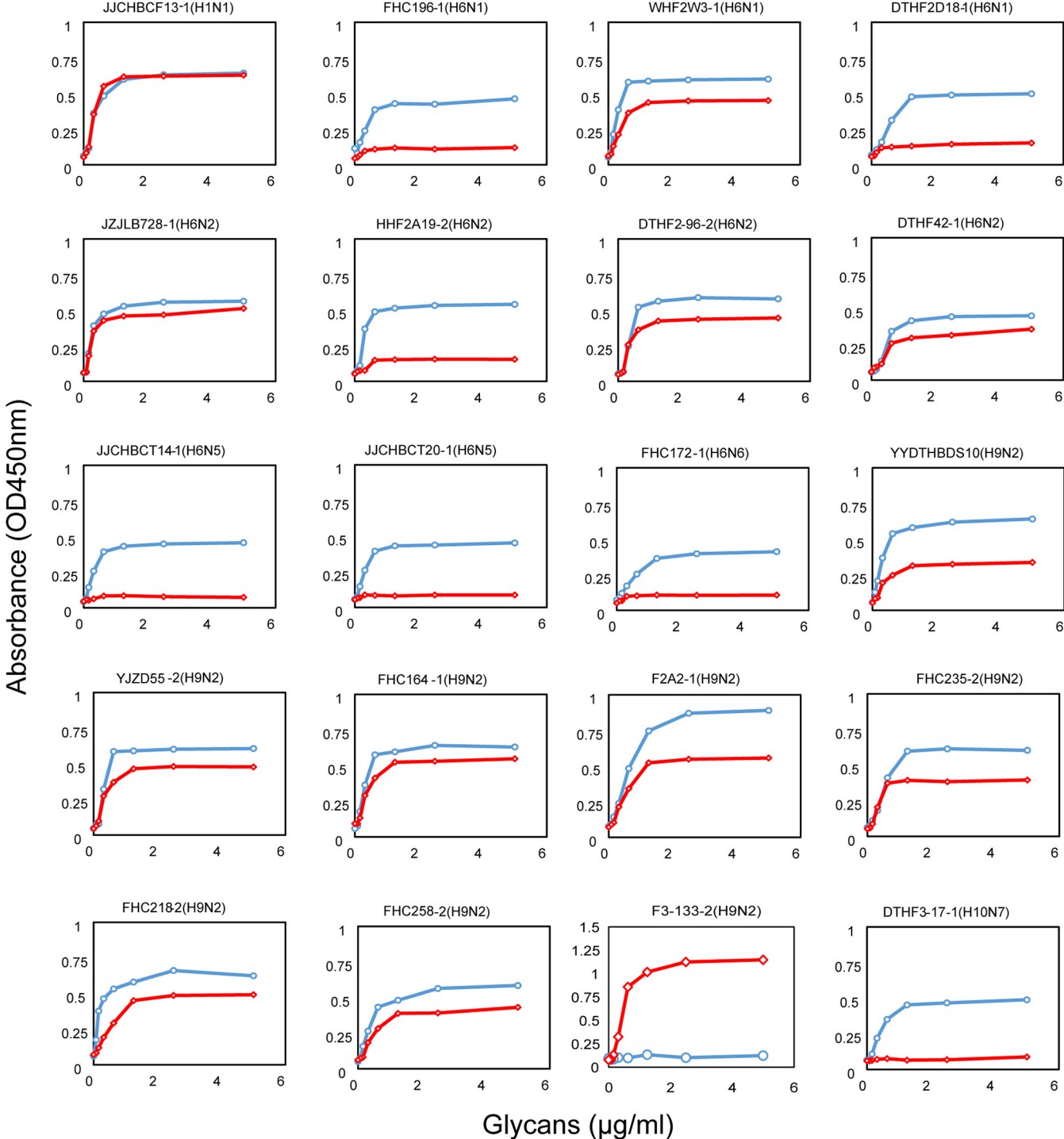

**FIG 5** Receptor-binding properties of the AIV isolates. Twenty viruses in this study that carried different key amino acids in receptor binding sites were chosen as representative strains (details are shown in Table S2 in the supplemental material). Receptor-binding properties of these viruses to human ($\alpha$2-6-SA) or avian ($\alpha$2-3-SA) receptors were tested using the solid-phase direct binding assay with trisaccharide receptors. Red and blue represent human- and avian-origin receptors, respectively. OD$_{450}$, optical density at 450 nm.

## DISCUSSION

We identified 12 subtypes of AIVs circulating in wild birds in the EA flyway over a period of 5 years (2015 to 2019). Overall, H9N2 was the dominant subtype, and the H6 subtype had a diverse NA combination. Reportedly, H9N2 has replaced H5N6 and H7N9 as the dominant AIV subtype in both chickens and ducks in live poultry markets

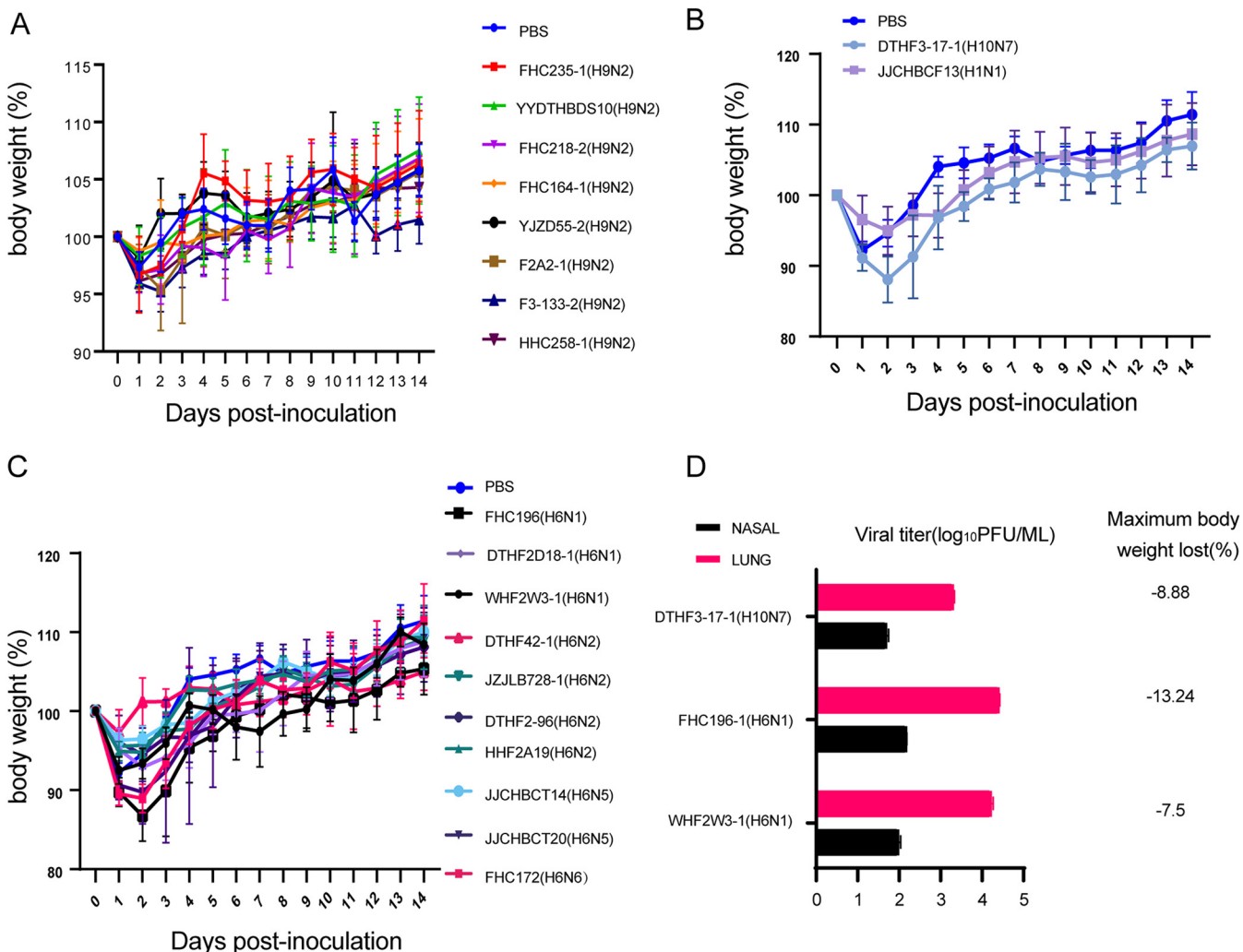

**FIG 6** Replication and pathogenicity of avian influenza viruses (AIVs) in mice. (A to C) Body weight loss curves of mice inoculated with H9 (A), H1 and H10 (B), or H6 (C) subtype viruses. Groups of mice ($n$ = 5) were intranasally infected with $10^6$ $EID_{50}$ in a 50-$\mu$L volume of each virus or phosphate-buffered saline (PBS). The body weight loss of the animals was monitored through 14 dpi Error bars represent the mean $\pm$ SD. (D) Viral titers (3 days postinfection [dpi]) in the lung and nasal passages of mice. Only viruses detected in both the lungs and nasal passages are shown; viral titers of all viruses tested in this study are detailed in Table 2.

(LPMs) (26). Wang et al. also showed that environmental samples that were collected from LPMs and wild bird habitats had significantly higher H9 positivity rates (27). H9N2 has become the dominant subtype in both wild birds and poultry, although the genome of H9N2 viruses in wild birds is distinct from that in poultry. Shi et al. (26) have reported that H9N2 acts as an incubator in LPMs by providing the internal genes to other virus subtypes, such as H7N9, H10N8, and H5N6. Whether H9N2 in wild birds can have the same role requires additional surveillance. Moreover, we detected two highly pathogenic viruses, H5N6 (clade 2.3.4.4h) and H5N8. A recent study shows that H5N6 viruses of clade 2.3.4.4h have become the dominant H5N6 viruses in poultry and wild birds since 2017 (28), and one isolate of them has caused one human infection case in March 2021 in Lao People's Democratic Republic (29). The H5N8 virus in this study was collected in late 2016, coinciding with the second wave of H5N8 outbreaks (2016 to 2017) (30). After that, HPAI H5N8 outbreaks frequently occurred in Europe and Asia from 2018 to early 2022 (31–33). Therefore, HPH5 viruses in this study are partly representative or indicative of the circulation of HPH5 viruses in Eurasia. Monitoring AIVs in wild birds plays an important role in the early warning system of AIV outbreaks.

**TABLE 2** Replication ability of avian influenza viruses in mice

| Virus | Replication of AIVs in mice[a] ($\log_{10}$ PFU/mL) (no. of AIV-positive samples/no. of total samples) | | Maximum body wt loss (%) |
|---|---|---|---|
| | Nasal | Lung | |
| JJCHBCF13-1 (H1N1) | / | 1.95 ± 0.1 (3/3) | −5.02 |
| DTHF2D18-1 (H6N1) | / | 2.48 ± 0.01 (3/3) | −7.12 |
| WHF2W3-1 (H6N1) | 1.98 ± 0.06 (3/3) | 4.22 ± 0.06 (3/3) | −7.5 |
| FHC196-1 (H6N1) | 2.17 ± 0.01 (3/3) | 4.40 ± 0.03 (3/3) | −13.24 |
| DTHF42-1 (H6N2) | / | 1.96 ± 0.12 (3/3) | −2.69 |
| HHF2A19-2 (H6N2) | / | 2.13 ± 0.01 (3/3) | −5.18 |
| JZJLB728-1 (H6N2) | / | 2.05 ± 0.02 (3/3) | −4.46 |
| DTHF2-96-2 (H6N2) | / | 1.97 ± 0.01 (3/3) | −3.94 |
| JJCHBCT14-1 (H6N5) | / | 2.04 ± 0.01 (3/3) | −3.68 |
| BCT20/2019 (H6N5) | / | 2.13 ± 0.01 (3/3) | −10.25 |
| FHC172 (H6N6) | / | 2.11 ± 0.04 (3/3) | −11.04 |
| FHC235-1 (H9N2) | / | 2.13 ± 0.02 (3/3) | −3.3 |
| YJZD55-2 (H9N2) | / | / | −1.91 |
| FHC218-2 (H9N2) | / | 1.69 ± 0.02 (2/3) | −2.86 |
| F3-133-2 (H9N2) | / | 2.60 ± 0.01 (3/3) | −4.77 |
| YYDTHBDS10 (H9N2) | / | 1.98 ± 0.01 (3/3) | −1.68 |
| F2A2-1 (H9N2) | / | 1.96 ± 0.01 (3/3) | −4.5 |
| FHC164-1 (H9N2) | / | 1.69 ± 0.13 (3/3) | −1.24 |
| HHC258-1 (H9N2) | / | 1.87 ± 0.04 (3/3) | −3.87 |
| DTHF3-17-1 (H10N7) | 1.70 ± 0.05 (3/3) | 3.30 ± 0.04 (3/3) | −8.88 |

[a]6-week-old BALB/c mice were inoculated with $10^{6.0}$ $EID_{50}$ of each virus in a 50-$\mu$L volume. Virus titers were shown as mean ± standard deviations. /, virus was not detected from the tissue.

Phylogenetic analysis of the isolates revealed that AIVs circulating in wild birds have various genotypes, with at least 31 internal gene constellations. Different AIV subtypes showed similar gene distribution, and the internal genes of the same AIV subtype showed high diversity, suggesting the occurrence of complex and extensive reassortment among AIVs circulating in wild birds. Moreover, the analysis of interseasonal persistence of internal genes revealed the prevalence of a portion of the AIV gene segment in several migration seasons, but the whole-genome persistence was rare (only occurring in H5N6 viruses). This further demonstrates that frequent reassortment occurred among the LPAIVs. Furthermore, 48 viruses, including H9N2, H1N1, H1N2, H6N1, H6N2, and H6N5, were found to carry genes from the North American lineage. In fact, a high proportion of wild bird AIVs exhibit evidence of intercontinental reassortment in both North America (34) and Europe (35). Intercontinental exchange of entire viruses was also detected in multiple AIVs (H2N5, H4N8, H3N6, H5N2, H6N5, H6N8, H16N3, and H9N2 subtypes) of both North American and Eurasian lineages (6, 13, 36). This corroborates extensive two-way gene or virus transmission between AIVs of the North American and Eurasian lineages.

Also, our results showed that gene diversity differed among segments and subtypes. The diversity in nucleotide sequences of each gene segment of AIVs from wild birds might reflect the breadth of sources from which the AIVs were derived (37). In this study, intercontinental gene segments were observed in H1, H6, and H9 AIVs. Therefore, the high gene diversity which some gene segments of these viruses present may partly result from the geographic separation of wild bird populations.

Furthermore, our results suggest a potential threat of AIVs in wild birds to human health. The whole genomes of one H6N6 virus were derived from poultry, indicating that spillover events occurred in LPAIs. Moreover, most viruses isolated in this study exhibited the ability to bind to human-type receptors. F3-133-2 (H9N2), in which the HA gene is derived from poultry lineage, binds exclusively to the $\alpha$-2,6-linked sialic acid receptor, as was observed with most of the H9N2 viruses circulating in LPMs in China (38). The remaining H9N2 viruses tested that could bind to both avian- and human-type receptors carried the HA-155T and HA-226Q substitutions. The I155T

mutation plays important roles in the binding of H9N2 virus to human-type receptors, but viruses with the 226Q mutation preferred avian-type receptors (39). Thus, we inferred that the two substitutions have opposite effects, leading to dual receptor-binding capacity of H9N2 viruses in wild birds. For the H6 viruses tested, all H6N5 and H6N6 viruses could bind only to $\alpha$-2,3-linked sialic acid receptors. One H6N1 virus and three H6N2 viruses acquired dual receptor-binding ability. We compared the HA gene sequences of these viruses with those of other H6 viruses and found that all four viruses carried the 160A substitution. In 2009, Chen et al. demonstrated that 160A mutation in the HA gene is pivotal for the H5N1 virus to bind to human-like receptors and to transmit in a mammalian host (17). Therefore, HA-160A may play a similar role in H6 viruses, but this needs to be verified experimentally.

The results of our *in vivo* experiments showed that most strains isolated from wild birds could replicate in mammals without prior adaptation, except for YJZD55-2 (H9N2), whose whole genome is derived from the North American lineage. Two H6N1 viruses and one H10N7 virus have a relatively higher adaptation to mice than the other viruses tested. No specific amino acids were identified to explain the difference in the replication capacity in mice. The molecular basis needs to be investigated in further studies. Body weight loss curves showed that mice infected with viruses in this study began to regain weight on the second or third days after infection. All mice eventually recovered their body weight, which indicated that viruses were cleared during the observation period (14 days). It must be pointed out that there is a limitation in this study: the viral titers of organs are only tested 3 days after infection, so it was impossible to know the exact time point of viral clearance and evaluate the time of viruses persisting in mice.

In conclusion, we found that the AIVs in wild birds undergo complex and frequent reassortment events. Through frequent gene flow between poultry and wild birds, in both the Western and Eastern hemispheres, AIVs acquiring new mutations or genotypes might become more virulent or more transmissible. Considerable numbers of AIVs in wild birds have acquired the ability to bind to human-type receptors and can replicate in mice. The potential for AIVs to spread globally via bird migration poses threats to human health worldwide. In view of these findings, it is necessary to increase AIV surveillance in wild birds and to continue evaluating the threat of these viruses to humans.

## MATERIALS AND METHODS

**Ethical approval.** The present study was carried out in strict accordance with the recommendations in the Guide for the Care and Use of Laboratory Animals of the Ministry of Science and Technology of the People's Republic of China. The study protocol was approved by the Committee on the Ethics of Animal Experiments of the Wuhan Institute of Virology, Chinese Academy of Science (approval no. WIVA04202001).

**Sample collection and virus isolation.** Fresh and well-separated wild bird feces were collected during every migration season (November of the first year to March of the following year) from November 2015 to March 2019. The fecal samples were collected using sterile cotton swabs, placed in viral transport medium, transported to the laboratory within 24 h at 4°C, and frozen at −80°C for future use. Viruses were isolated using 10-day-old specific-pathogen-free (SPF) chicken embryos according to a World Health Organization (WHO) manual (40). The allantoic fluid of the inoculated eggs was collected and tested for the presence of hemagglutinin activity. Hemagglutinin-positive samples were further tested by reverse transcriptase PCR (RT-PCR) using universal primers targeting the M gene as described previously (41).

**RNA extraction, RT-PCR, and subtyping.** Using a nucleic acid extraction system with matched EX-RNA/DNA viral nucleic acid extraction kits (Tianlong Science and Technology, Co., Ltd.), RNA was extracted from AIV-positive allantoic fluid. The RNA was reverse transcribed into single-stranded DNA using Moloney murine leukemia virus (M-MLV) reverse transcriptase (Promega, Madison, USA). PCR was performed for HA and NA subtyping with the primers described previously (41).

**Next-generation sequencing.** Next-generation sequencing (NGS) was used to determine the whole-genome sequences of AIV isolates. The libraries were prepared and sequenced on the Illumina HiSeq 4000 platform as described previously (42). Briefly, viral RNA was extracted and reverse transcribed to cDNA. After library preparation, 250-bp paired-end sequencing was conducted. The sequencing depth for AIV isolates was 0.2 Gb per sample.

**Sequencing data filter and assembly.** NGS reads were processed by filtering out low-quality reads (≥10 bases with qualities < 10), adaptor-contaminated reads (with >15 bp matched to the adapter

sequence), poly-nucleotides (≥8 Ns), duplication, and host-contaminated reads (soap2 version 2.21; <5 mismatches) (43). The filtered reads were assembled by mapping to the Influenza database (44), and the best-matched reference sequences were selected. MAQ software (version 0.6.6) was used to perform reference-based assembly (45).

**Phylogenetic analyses.** All influenza A virus sequences submitted as of November 2019 were retrieved from the Influenza Virus Resource at the National Center for Biotechnology Information (NCBI), according to the *BLAST Command Line Applications User Manual* provided by NCBI (46). Multiple-sequence alignments were generated using MAFFT version 7 and manually edited using BioEdit version 7.1.3.0 (47). Only the coding DNA sequence (CDS) region was retained for phylogenetic analysis. All phylogenetic trees were constructed using the IQ-TREE software (48), and 1,000 bootstrap replicates for each tree were performed. Phylogenetic analyses were conducted over two rounds: in the first round, 100 of the most homologous sequences of each gene were used to infer the overall topology; in the second round, based on the phylogenetic analysis, only representative sequences of each cluster were retained.

The Bayesian Markov chain Monte Carlo (BMCMC) method in BEAST version 1.8.3 (49) was used to generate the maximum clade credibility (MCC) trees of 22 H9N2 viruses harboring North American lineage genes. We blasted the top 50 hits of each gene segment of the 22 viruses to build a data set. Temporal outliers possibly indicative of sequencing errors were removed by inspecting the correlation coefficient between regressions of root-to-tip divergence and sampling times by using TempEst version 1.5.1 (50). Correlation coefficients were >0.6 for all genes. Each gene segment data set was analyzed using the codon-based SRD06 nucleotide substitution (51) and constant-coalescent population models with a strict clock and chain length of 100,000,000 steps. Convergence of the runs was confirmed using Tracer version 1.7 (http://tree.bio.ed.ac.uk/software/tracer/), and effective sample size values of 200 indicated a sufficient level of sampling. All MCC trees were summarized using TreeAnnotator with 10% burn-in cutoffs and visualized using FigTree version 1.4.3 (http://tree.bio.ed.ac.uk/software/figtree/).

**Quantification of nucleotide sequence diversity.** Complete alignments of each internal gene were performed using the PopGenome package in R (version 3.4.2) to estimate nucleotide sequence diversity (52). Per-site diversity was calculated by dividing the nucleotide sequence diversity output by the number of sites present in each alignment. As each subset contained different numbers of sequences, this value was normalized by dividing by the number of sequences in each respective data set. Heatmaps from these data were plotted in R (version 3.4.2).

**Receptor-binding assay.** Receptor-binding specificity was determined using the solid-phase direct binding assay as described previously (53). Briefly, 96-well microtiter plates were coated with biotinylated glycan $\alpha$2-3-SA receptor (trisaccharide, Neu5Ac$\alpha$2-3Gal$\beta$1-4GlcNAc$\beta$-SpNH-LC-LC-biotin, and pentasaccharide, NeuAc$\alpha$2-3Gal$\beta$1-4GlcNAc$\beta$1-3Gal$\beta$1-4GlcNAc$\beta$1-SpNH-LC-LC-biotin) and $\alpha$2-6-SA receptor (trisaccharide, Neu5Ac$\alpha$2-6Gal$\beta$1-4GlcNAc$\beta$-SpNH-LC-LC-biotin, and pentasaccharide, NeuAc$\alpha$2-6Gal$\beta$1-4GlcNAc$\beta$1-3Gal$\beta$1-4GlcNAc$\beta$1-SpNH-LC-LC-biotin). Virus dilutions containing 64 HA units were incubated with NA inhibitors (NAIs) (10 $\mu$M [each] oseltamivir and zanamivir). Virus-receptor binding was determined using rabbit antisera against influenza viruses (H1, H6, H9, and H10) and horseradish peroxidase (HRP)-linked goat anti-rabbit antibodies (Bioeasytech, Beijing, China). Using tetramethylbenzidine as theF substrate, the results were measured at 450 nm.

**Virus infection in mice.** Groups of eight 6-week-old female BALB/c mice (Vital River Laboratories, Beijing, China) were lightly anesthetized with $CO_2$ and inoculated intranasally (i.n.) with 106 50% egg-infective dose ($EID_{50}$) of AIV at a volume of 50 $\mu$L. Control mice were inoculated with 50 $\mu$L of phosphate-buffered saline. At 3 days postinfection (dpi), three of the eight inoculated mice in each group were euthanized, and the lungs, nasal turbinates, kidneys, liver, spleen, and brain were collected for virus titration in MDCK cells. The remaining mice (five in each group) were monitored daily for weight loss and survival for 14 days.

**Data availability.** The whole-genome sequences of all viruses reported in this paper have been deposited into GISAID (https://www.gisaid.org/). The accession numbers of gene sequences are provided in Table S2.

## SUPPLEMENTAL MATERIAL

Supplemental material is available online only.
**SUPPLEMENTAL FILE 1**, XLSX file, 0.01 MB.
**SUPPLEMENTAL FILE 2**, XLSX file, 0.04 MB.
**SUPPLEMENTAL FILE 3**, PDF file, 3 MB.

## ACKNOWLEDGMENTS

We are grateful to the Core Facility and Technical Support and biosafety level 3 (BSL-3) laboratory of Wuhan Institute of Virology, CAS, and the National Field Scientific Observation and Research Station of Dongting Lake Wetland Ecosystem in Hunan Province. We thank the authors and submitting laboratories of these sequences from the Global Initiative on Sharing All Influenza Data (GISAID) EpiFlu Database (www.gisaid.org) and National Center for Biotechnology Information (NCBI) Influenza Virus Database (https://www.ncbi.nlm.nih.gov/). We thank Editage (www.editage.cn) for English language editing.

This work was supported by the National Science and Technology Major Projects (grant numbers 2020ZX10001016 and 2018ZX10101004). The funders had no role in study design, data collection and analysis, decision to publish, or preparation of the manuscript.

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
