## [Reviewer comments · Microbiology Spectrum]

Microbiology Spectrum

Genetic and pathogenic characterization of Avian influenza virus in migratory birds between 2015 and 2019 in Central China

Quanjiao Chen, Zhongzi Yao, Huabin Zheng, Jiasong Xiong, Liping Ma, Rui Gui, Gongliang Zhu, Yong Li, Guoxiang Yang, Guang Chen, and Jun Zhang

Corresponding Author(s): Quanjiao Chen, CAS Key Laboratory of Special Pathogens and Biosafety, CAS Center for Influenza Research and Early Warning, Wuhan Institute of Virology, Chinese Academy of Sciences

Review Timeline:

Submission Date:	May 4, 2022
Editorial Decision:	May 26, 2022
Revision Received:	June 10, 2022
Accepted:	June 22, 2022

Editor: Heba Mostafa

Reviewer(s): The reviewers have opted to remain anonymous.

Transaction Report:

DOI: <https://doi.org/10.1128/spectrum.01652-22>

May 26, 2022

Prof. Quanjiao Chen
CAS Key Laboratory of Special Pathogens and Biosafety, CAS Center for Influenza Research and Early Warning, Wuhan
Institute of Virology, Chinese Academy of Sciences
xiaohongshan 44#, wuchang
Wuhan, Hubei 430071
China

Re: Spectrum01652-22 (Genetic and pathogenic characterization of Avian influenza virus in migratory birds between 2015 and 2019 in Central China)

Dear Prof. Quanjiao Chen:

Link Not Available

Sincerely,

Heba Mostafa

Journals Department
Reviewer comments:

Reviewer #1 (Comments for the Author):

Active surveillance of circulating influenza viruses is essential to anticipate spillover events in other species and prepare for possible pandemics. Yao et al present here a characterization of influenza viruses in wild birds during 4 migratory seasons from 2015 to 2019 in central China. They find frequent gene reassortments between subtypes and North America/Eurasia origin and identified 2 spillover events from poultry. Importantly, they detected 2 highly pathogenic virus and IAV of H1, H6, H9 and H10 subtypes with the ability to bind human receptors and replicate in mice. The study of active surveillance provides important information of the circulating IAV in migratory birds. The manuscript is well

written and rigorous analyses are presented. A few points listed below need clarification and/or correction.

Comments:

1. The pathogenicity characterization of the viruses is limited. Some controls are missing (especially the PBS control in Fig 6) and no statistics is done. Moreover, the viruses were recovered from mice at day 3 post infection but because there are no further time points, it is unknown whether they can actively replicate and grow in mammals. Standard viruses could have been used as controls to better conclude the experiments. Some precautions should be taken when concluding the results and the limitation of this experiment clearly mentioned.

How to explain the H10N7 virus showed avian receptor binding specificity only but was recovered in mice in nasal and lungs?

2. There is no data regarding recent migratory seasons post 2019. I understand those data may not be processed yet in the area. Brief surveillance reports are published regularly and could be used here to discuss the data. Can the authors put into perspective their results with other available surveillance data in mainland China (H Bo, ...D-Y Wang, 2021 for example) and Europe/North America?

Other specific comments:

3. Fig 1: can the authors indicate the number of total samples for each migratory season. A legend for the x axis is needed in panel C.

4. Fig 6 and Table 2: The PBS control is missing on Fig 6 panel B and C and should be added. Please indicate what the results show and what error bars correspond to (Mean +/- SEM or SD...). In Table 2, it would be helpful to see in an additional column in how many animals the virus was detected.

5. L69-70: please edit the sentence "H10N8 viruses were detected...and harbored..."

6. L712-713: Please rewrite the sentence.

7. L719: (C) show H6 subtype and not H10. Please correct.

Reviewer #2 (Comments for the Author):

This manuscript describes a five-year active surveillance in wild birds in Central China, along the East Asian-Australasian flyway, from 2015 to 2019. Twelve HA-NA subtype combinations were detected. Phylogenetic analysis revealed intercontinental transmission, continuous reassortment events. The manuscript provides valuable information on the genetic and biological properties of AIV from wild birds in China from 2015 to 2019. These data are important to understand the evolution, intercontinental spread and zoonotic potential of wild-bird AIV.

The following comments should be addressed:

Major comments:

1. it is not clear why this number of samples was collected. How the authors determined the sample size for their systemic study? The variable number of samples for each season may be the cause for the variable prevalence rate and subsequently the distribution of different subtypes per season. The authors should explain the reasons for these variations.

2. The assumption that a virus has zoonotic potential based on binding affinity to small glycan can be misleading. Replication kinetics of selected viruses (e.g. with high replication efficiency in mice) in human cells should be done.

3. Line 383: How did the author distinguish the droppings of free-range poultry from wild birds and identify the bird species?

Minor comments:

1. Line 85: More details on the EA flyway (e.g. no. of birds, destinations, time of migration, most common species, etc.) should be added in the introduction section to highlight the importance of this study.

2. Lines 153-159: it would be helpful to discuss these results in the discussion section.

3. Line 278: "...total of 20 representative strains", these AIV should be marked in the phylogenetic tree (at least in the supplementary figures).

Staff Comments:

Preparing Revision Guidelines

To submit your modified manuscript, log onto the eJP submission site at <https://spectrum.msubmit.net/cgi-bin/main.plex>. Go to Author Tasks and click the appropriate manuscript title to begin the revision process. The information that you entered when you first submitted the paper will be displayed. Please update the information as necessary. Here are a few examples of required

updates that authors must address:

Please return the manuscript within 60 days; if you cannot complete the modification within this time period, please contact me. If you do not wish to modify the manuscript and prefer to submit it to another journal, please notify me of your decision immediately so that the manuscript may be formally withdrawn from consideration by Microbiology Spectrum.

Response to Reviewers

Reviewer(s)' Comments to Author:

Reviewer: 1

Active surveillance of circulating influenza viruses is essential to anticipate spillover events in other species and prepare for possible pandemics. Yao et al present here a characterization of influenza viruses in wild birds during 4 migratory seasons from 2015 to 2019 in central China. They find frequent gene reassortments between subtypes and North America/Eurasia origin and identified 2 spillover events from poultry. Importantly, they detected 2 highly pathogenic virus and IAV of H1, H6, H9 and H10 subtypes with the ability to bind human receptors and replicate in mice.

The study of active surveillance provides important information of the circulating IAV in migratory birds. The manuscript is well written and rigorous analyses are presented. A few points listed below need clarification and/or correction.

Q1: The pathogenicity characterization of the viruses is limited. Some controls are missing (especially the PBS control in Fig 6) and no statistics is done. Moreover, the viruses were recovered from mice at day 3 post infection but because there are no further time points, it is

unknown whether they can actively replicate and grow in mammals.

Standard viruses could have been used as controls to better conclude the experiments. Some precautions should be taken when concluding the results and the limitation of this experiment clearly mentioned.

How to explain the H10N7 virus showed avian receptor binding specificity only but was recovered in mice in nasal and lungs?

A1: Thanks for your advice. We have added the PBS control in Fig 6 B and C panel, and annotated the error bars correspond to standard deviation in Figure legends (line 764-767 in the revised manuscript) (the revised manuscript means clean version rather than track version). We agree that further time points would more accurately reflect whether viruses can actively replicate and grow in mice, and the limitation of this experiment does exist. It is impossible to know the exact time point of viral clearance and evaluate the duration of different viruses persisting in mice. However, the body weight loss curves of mice infected with viruses showed that all mice eventually recovered their body weight, which could indicate the viruses were cleared during observation period (14 days).

We agree with you that it's better to test the viral titers of the organs at more time points and use standard viruses as control to get more meaningful and rigorous conclusions. We'll improve the experimental

design in the future, and we have discussed the limitations in line 393-397 in the revised manuscript.

The α -2,3-linked receptors can be expressed in the lower respiratory tract and lungs of both human and mice, so some avian influenza viruses can replicate in the lungs of mammals but lack infectivity (*Nature*. 2006 Mar 23;440(7083):435-6.). The sialic acid receptor spectrum in mice is slightly different from that in humans. α -2,3 and α -2,6-linked sialic acid receptors are proved to be expressed both in the basal and connective tissue of the nasal of mice (*Vet Res Commun*. 2009 Dec;33(8):895-903.), which could explain the H10N7 virus showed avian receptor binding specificity only but was recovered in mice in nasal and lungs.

Q2: There is no data regarding recent migratory seasons post 2019. I understand those data may not be processed yet in the area. Brief surveillance reports are published regularly and could be used here to discuss the data. Can the authors put into perspective their results with other available surveillance data in mainland China (H Bo, ...D-Y Wang, 2021 for example) and Europe/North America?

A2: We have discussed other surveillance data together with our results in line 316-318 and line 323-334 in the revised manuscript.

Q3: Fig 1: can the authors indicate the number of total samples for each migratory season. A legend for the x axis is needed in panel C.

A3: We have indicated the number of total samples for each migratory season in panel B and added the legend of x axis in panel C in Fig 1.

Q4: Fig 6 and Table 2: The PBS control is missing on Fig 6 panel B and C and should be added. Please indicate what the results show and what error bars correspond to (Mean +/- SEM or SD...). In Table 2, it would be helpful to see in an additional column in how many animals the virus was detected.

A4: We have modified the Fig 6 (include the legend of Fig 6 in line 764-767 in revised manuscript) and Table 2 according to your advice.

Q5:L69-70: please edit the sentence "H10N8 viruses were detected...and harbored..."

A5: We have edited this sentence in line 69-72 in the revised manuscript.

Q6: L712-713: Please rewrite the sentence.

A6: We have rewritten this sentence in line 757 in the revised manuscript.

Reviewer: 2

This manuscript describes a five-year active surveillance in wild birds in Central China, along the East Asian-Australasian flyway, from 2015 to 2019. Twelve HA-NA subtype combinations were detected. Phylogenetic analysis revealed intercontinental transmission, continuous reassortment events. The manuscript provides valuable information on the genetic and biological properties of AIV from wild birds in China from 2015 to 2019. These data are important to understand the evolution, intercontinental spread and zoonotic potential of wild-bird AIV.

The following comments should be addressed:

Q1: 1.it is not clear why this number of samples was collected. How the authors determined the sample size for their systemic study? The variable number of samples for each season may be the cause for the variable prevalence rate and subsequently the distribution of different subtypes per season. The authors should explain the reasons for these variations.

A1: Most of sampling sites in this study are located in National Nature Reserve, where specific permission is required to enter. The open area for sampling is limited. Therefore, it is hard to guarantee that enough samples are collected in every sampling activity, and we just try to get as many fresh samples as we can. This is the reason why the sample size is

variable among seasons. However, we sampled monthly in the migration seasons to avoid the sampling bias.

Q2: The assumption that a virus has zoonotic potential based on binding affinity to small glycan can be misleading. Replication kinetics of selected viruses (e.g. with high replication efficiency in mice) in human cells should be done.

A2: Yes, we agree with you. Evaluating a virus has zoonotic potential based on binding affinity to small glycan can be misleading. So we tested the replication ability of the viruses in mice. Our data showed that most viruses (19/20) can replicate in mice without prior adaptation, and we think the data in mice infection are more powerful than replication kinetics of viruses in human cells.

Q3: How did the author distinguish the droppings of free-range poultry from wild birds and identify the bird species?

A3: The sampling sites is in the nature reserve, where free-range poultry is hard to access. And the shape of most fecal samples collected by us were quite different from poultry (as shown in the picture below).

Fecal samples in this study

Q4: Line 85: More details on the EA flyway (e.g. no. of birds, destinations, time of migration, most common species, etc.) should be added in the introduction section to highlight the importance of this study.

A4: We have added more information about EA flyway in line 82-89 in the revised manuscript.

Q5: Lines 153-159: it would be helpful to discuss these results in the discussion section.

A5: We have added related content in line 354-360 in the revised manuscript.

Q6: Line 278: "..total of 20 representative strains", these AIV should be marked in the phylogenetic tree (at least in the supplementary figures).

A6: We have marked the representative strains in supplementary Table 2.

June 22, 2022

Prof. Quanjiao Chen
CAS Key Laboratory of Special Pathogens and Biosafety, CAS Center for Influenza Research and Early Warning, Wuhan
Institute of Virology, Chinese Academy of Sciences
xiaohongshan 44#, wuchang
Wuhan, Hubei 430071
China

Re: Spectrum01652-22R1 (Genetic and pathogenic characterization of Avian influenza virus in migratory birds between 2015 and 2019 in Central China)

Dear Prof. Quanjiao Chen:

Your manuscript has been accepted, and I am forwarding it to the ASM Journals Department for publication. You will be notified when your proofs are ready to be viewed.

Sincerely,

Heba Mostafa
Editor, Microbiology Spectrum

Journals Department
Supplemental Table 1: Accept
Supplemental Material: Accept
Supplemental Table 2: Accept